# One-bit Deep Hashing: Towards Resource-Efficient Hashing Model with Binary Neural Network

## ABSTRACT

Deep Hashing (DH) has emerged as an indispensable technique for fast image search in recent years. However, using full-precision Convolutional Neural Networks (CNN) in DH makes it challenging to deploy on devices with limited resources. To deploy DH on resource-limited devices, the Binary Neural Network (BNN) offers a solution that significantly reduces computations and parameters compared to CNN. Unfortunately, applying BNN directly to DH will lead to huge performance degradation. To tackle this problem, we first conducted extensive experiments and discovered that the center-based method provides a fundamental guarantee for BNN-DH performance. Subsequently, we delved deeper into the impact of BNNs on center-based methods and revealed two key insights. First, we find reducing the distance between hash codes and hash centers is challenging for BNN-DH compared to CNN-based DH. This can be attributed to the limited representation capability of BNN. Second, the evolution of hash code aggregation undergoes two stages in BNN-DH, which is different from CNN-based DH. Thus, we need to take into account the changing trends in code aggregation at different stages. Based on these findings, we designed a strong and general method called One-bit Deep Hashing (ODH). First, ODH incorporates a semantic self-adaptive hash center module to address the problem of hash codes inadequately converging to their hash centers. Then, it employs a novel two-stage training method to consider the evolution of hash code aggregation. Finally, extensive experiments on two datasets demonstrate that ODH can achieve significant superiority over other BNN-DH models. The code for ODH is available at https://anonymous.4open.science/r/OSH-1730.

## CCS CONCEPTS

• **Information systems** → **Learning to rank**; *Similarity measures*.

## KEYWORDS

deep hashing; binary neural network; image retrieval

## 1 INTRODUCTION

Hashing has played a crucial role in image retrieval in the multimedia community, which aims to return the most relevant images from the database according to a given image query [19]. This approach represents images as binary vectors, also known as hash codes. Driven by the storage and search efficiency offered

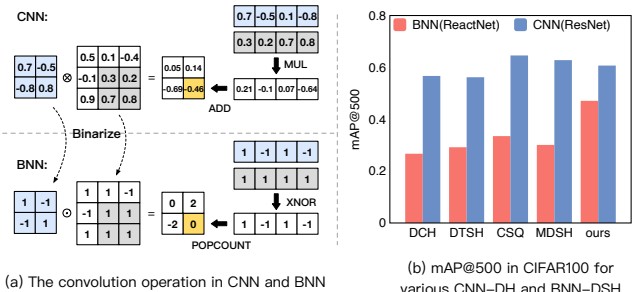

(a) The convolution operation in CNN and BNN

(b) mAP@500 in CIFAR100 for various CNN–DH and BNN–DSH

**Figure 1: (a) The convolution operation procedure in CNN and BNN. Compared to CNN, BNN offers more efficient operation and storage. (b) The $mAP@500$ performance on the CIFAR100 dataset when deep hashing models adopt CNN or BNN as the backbone.**

by hash codes, hashing is an active research topic in image retrieval [2, 3, 6, 8, 10, 15, 16, 27, 29–33, 40, 44]. In recent years, deep Convolutional Neural Networks (CNN) have shown remarkable performance in learning a mapping from images to hash codes for hashing, known as Deep Hashing (DH). However, the noteworthy performance achieved by CNN-based Deep Hashing (CNN-DH) often entails a trade-off in terms of larger model size and increased computational complexity [41, 42]. It hinders the application of DH on resource-limited devices. For example, many handheld devices and small drones lack the necessary GPUs and sufficient memory to accommodate computationally intensive CNN models. Therefore, it is crucial to reduce the computational and memory requirements of DH largely while preserving its performance.

Towards this goal, Binary Neural Networks (BNNs) have emerged as a promising approach for compressing neural networks [25, 39]. As depicted in Figure 1 (a), BNNs aim to reduce storage and computation costs by binarizing both weights and activations. This characteristic enables BNNs to achieve a memory compression ratio of 32× and computational reductions of up to 58× on specialized processors [24]. BNNs' inherent compatibility with DH manifests through the XNOR and POPCOUNT operations employed in BNN convolutions align harmoniously with the Hamming distance calculation on hash codes. Besides, in recent years, the advanced BNNs [34, 35] have rapidly narrowed the performance gap with CNN in various tasks [21, 22, 36]. Recognizing the immense value of BNNs in facilitating the deployment of DH on resource-limited devices, the quest for an effective BNN-based DH (BNN-DH) holds excellent practical significance.

Unfortunately, when we replace the original CNN backbone with a BNN backbone in DH, we observe a significant performance degradation as illustrated in Figure 1 (b). This observation indicates that the current DH is not particularly well-suited for a BNN backbone. To tackle this problem, we initially conduct preliminary experiments, which reveal that center-based deep hashing methods

[6, 29, 30, 40] offer an essential foundation for the performance of BNN-DH. To further explore the bottlenecks that impact the performance of BNN-DH in center-based methods, we conduct two analyses from different perspectives: the optimization perspective and the hash code distribution perspective. (i) From the optimization perspective, center-based methods aim to bring the hash codes closer to their hash centers [40]. Thus, we conduct a distance analysis as shown in Figure 2, which reveals that drawing the hash codes closer to their hash centers becomes increasingly challenging in BNN-DH compared to CNN-DH. This phenomenon can be attributed to the limited representation capability of BNN, thereby impeding the attainment of optimization efficacy comparable to that of CNN-DH. (ii) From the hash code distribution perspective, center-based methods aim to force hash codes of similar images to aggregation. To explore this aspect, we conduct a hash code aggregation analysis. As depicted in Figure 3, the CNN-DH focuses on enhancing code aggregation. However, in BNN-DH, the evolution comprises two stages: space exploration and code aggregation. During the space exploration stage, hash codes with the same category tend to diverge from one another. In the subsequent code aggregation stage, hash codes with the same category start to approach one another laboriously. Therefore, it is necessary to consider the distinctive characteristics of code aggregation within the BNN-DH training process to avoid sub-optimal results.

Motivated by these analyses, we have developed a strong and general model called One-bit Deep Hashing (ODH) to tackle the aforementioned problems. First, we introduce a semantic self-adaptive hash center module to overcome the issue of hash codes not adequately approaching the their hash centers. It directly incorporates semantic information of categories during the hash center learning process, which makes the hash centers dynamically adjusted based on the current distribution of hash codes in the Hamming space. Second, we introduce a two-stage training method to align with the changing characteristics of code aggregation in BNN-DH. During the space exploration phase, we utilize dynamic hash centers as a global similarity metric and use a loss function to push the hash code close to its hash center but distanced from other hash centers. In the subsequent code aggregation stage, we include an additional aggregation extrapolation objective to enhance hash code aggregation. In summary, the paper makes three main contributions:

- We propose a novel model, One-bit Deep Hashing (ODH), which combines Binary Neural Networks (BNNs) and deep hashing (DH) into a unified framework. Our work is timely as CNN backbones' computational and memory requirements pose challenges for deploying DH on resource-limited devices. By leveraging BNNs, we enable DH to overcome this dilemma.
- Recognizing that BNN-DH exhibits distinct properties compared to its CNN-DH, we conducted some analyses to explore the inherent problems. Based on the insights gained from these analyses, we introduce a semantic self-adaptive hash center module and employ a two-stage training method. These techniques effectively bridge the gap between BNNs and CNNs in the context of deep hashing.
- We perform comprehensive experiments on two widely used datasets to evaluate the performance of ODH against other

baseline methods. The results demonstrate the superiority of ODH and establish it as a robust foundational model for future binarized deep hashing research, with implications for both academia and industry.

## 2 RELATED WORK

### 2.1 Deep Hashing

Deep Hashing (DH) has gained increasing importance in large-scale image retrieval tasks. It aims to convert images into hash codes using a deep neural network as the backbone and can be broadly categorized into supervised and unsupervised [19]. In this paper, we focus on supervised deep hashing techniques, which can be further subcategorized into pairwise methods, ranking-based methods, and center-based methods.

Pair-wise methods [3, 15, 27, 32, 33, 44] have been extensively studied and involve using pairwise similarities between data pairs as learning targets. The objective is to ensure that similar pairs of images have similar hash codes while dissimilar pairs have dissimilar hash codes. Additionally, various researchers have adopted ranking-based similarity preserving loss terms [2, 8, 16, 31]. For instance, triplet loss [16, 31] and list-wise loss [2, 8] are commonly used to maintain data ordering. However, our preliminary experiments find that pairwise and ranking-based methods make it hard to achieve satisfactory results across various BNN backbones.

Center-based methods [6, 10, 29, 30, 40] stem from the point-wise method [19], which has gained popularity recently. These methods first generate hash centers for each category. Then, they force hash codes outputted from the network to approach their hash centers. For example, CSQ [40] generates hash centers using the Hadamard matrix and Bernoulli sampling, and then it applies a central similarity objective to push hash codes approach hash centers. Subsequent works [6, 10, 29, 30] have expanded upon CSQ by elaborating on the objective functions [6, 10] or methodologies for generating hash centers [29, 30]. These methods generate fixed hash centers for image retrieval tasks, and hash codes can be easily close to the hash center when using full-precision CNN as shown in Figure 2, thus achieving excellent performance. However, our preliminary experiments show that when using BNN as the backbone, hash codes make it difficult to approach the hash center. Therefore, we consider making the hash centers movable to approach hash codes. One relevant work that could be referenced towards this goal is [12], yet it overlooks the semantic information of categories. Recent advancements in cross-modal hashing [5, 28, 38] propose the implicit integration of label semantic information into hash centers. Drawing inspiration from these works, we employ label vectors, albeit solely as indices for categories, and directly incorporate the semantic information of these categories to derive more precise hash centers.

### 2.2 Binary Neural Network

The Binary Neural Network (BNN) is considered the most efficient quantized network for resource-limited devices, as it quantizes full-precision weights and activations to binary values [25, 39]. Given a CNN, we simply denote its r-th layer real-valued weights as $W^r$ and the inputs as $A^r$. Then, BNN binarizes each weight $w^r \in W^r$ and each activation $a^r \in A^r$ to binary value $\{-1, +1\}$. The basic

binarization can be achieved by the sign function as follows:

$$Q(x^r) = sign(x^r) = \begin{cases} +1, & if\ x^r \geq 0 \\ -1, & otherwise, \end{cases} \quad (1)$$

where $Q(\cdot)$ denotes the quantization function. Then, efficient bit-wise XNOR and POPCOUNT operations can be used to conduct binarized convolution $\odot$ as follows:

$$Q(A^r) \odot Q(W^r) = POPCOUNT(XNOR(Q(A^r), Q(W^r))). \quad (2)$$

To enhance BNN performance, several notable efforts have been made. XnorNet [24] improves convolution efficiency by binarizing the weights and inputs of convolution kernels. Bi2Real [20] rescales feature maps based on input before binarized operations and incorporates a gating module similar to SE-Net [11]. ReActNet [17] replaces conventional PReLU and sign functions with RPReLU and RSign, respectively, using a learnable threshold to improve BNN performance. Rbonn [35] introduces a recurrent bilinear optimization to address the asynchronous convergence problem in BNNs. However, many of these methods suffer from weight oscillation caused by the non-parametric scaling factor. Rebnn [34] mitigates frequent oscillation to enhance BNN training.

Recently, two methods [41, 42] have explored the combination of BNN and DH for image retrieval. REDH [42] proposes regularizations on the binarization functions of weights and activations. BNNH [41] constructs a binarized network architecture to generate binary outputs directly and introduces an activation-aware loss to guide the update of activations in intermediate layers. However, these methods design specialized modifications or constraints in BNN architectures, occasionally constraining their applicability to other innovative BNNs. Our work aims to develop a deep hashing model that can be generalized to various types of BNNs.

## 3 PRELIMINARIES

### 3.1 Deep Hashing

Consider a database $X = \{x_1, ..., x_N\}$ comprising $N$ images. Deep Hashing (DH) targets to learn a hash function $f : x_i \mapsto h_i$ that maps each image $x_i \in X$ to a low-dimensional binary vector $h_i \in \{-1, 1\}^b$, also known as hash code, where $b$ denotes the length of hash code. The hash function $f$ usually utilizes a convolutional neural network (CNN) as the feature extractor. This mapping aims to preserve the pairwise similarities between the images $x_i$ and $x_j$ in the Hamming space, characterized by the Hamming distance $D_H(h_i, h_j)$ for hash codes $h_i$ and $h_j$. The Hamming distance between two hash codes is defined as the number of differing bits between the codes:

$$D_H(h_i, h_j) = \sum_{k=1}^{N} 1_{h_{ik} \neq h_{jk}} = POPCOUNT(h_i\ XNOR\ h_j), \quad (3)$$

where the summation can be computed efficiently due to the XNOR and POPCOUNT instruction that counts the number of bits set to one within a machine word [7].

### 3.2 Center-based Method

Center-based methods [6, 10, 29, 30, 40] have achieved remarkable performance in recent years. They consist of two phases. Taking the CSQ [40] as an example, in the first phase, hash centers $C =$

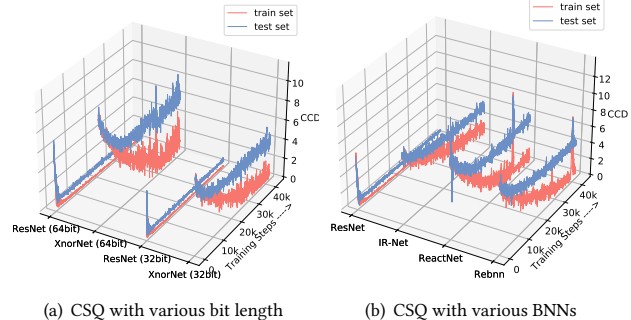

(a) CSQ with various bit length    (b) CSQ with various BNNs

**Figure 2: The Code Center Distance (CCD) analysis on the IM-AGENET100 dataset. A lower CCD indicates a closer distance between the hash codes and their hash centers. (a) CSQ employs ResNet50 or XnorNet as the backbone. (b) CSQ employs IR-Net, ReactNet, or Rebnn as the backbone. This analysis reveals that BNN-DH is struggling to achieve the same level of optimization as that of CNN-DH.**

$\{c_i\}_{i=1}^m, c_i \in \{-1, 1\}^b$ are generated. They are defined as follows:

$$\frac{1}{P} \sum_{i \neq j}^{m} D_H(c_i, c_j) \geq \frac{b}{2}, \quad (4)$$

where $m$ is the number of hash centers (typically equivalent to the number of categories) and $P$ is the number of combinations of different $c_i$ and $c_j$ in $C$. Hash centers represent a distributed learning target for different categories of images. CSQ [40] proposes two methods to generate hash centers, including sampling from a hadamard matrix or a bernoulli distribution. In other center-based methods [29, 30], various settings and methods may be proposed to generate hash centers. In the second phase, CSQ employs a binary cross-entropy loss to drive the hash code $h_i$ toward its hash center $c_i'$. While other center-based methods [10, 30] may employ different objectives, their primary goal remains consistent.

### 3.3 Preliminary Analysis

We first conduct a preliminary experiment as shown in Table 1 and reveal that center-based methods are more suitable to integrate with BNNs. To further explore the impact of BNNs on center-based methods, we conduct two analyses from the optimization and hash code distribution perspectives.

From the optimization perspective, center-based methods are to encourage the hash codes to approach their hash centers. To evaluate the distance between hash codes and their hash centers, we introduce the Codes Center Distance (CCD) metric defined as follows:

$$CCD = \frac{1}{N} \sum_{i=1}^{N} D_H(h_i, c_i'), \quad (5)$$

A lower value of $CCD$ indicates a closer distance between the hash codes and their hash centers. Figure 2 (a) illustrates the evolution of $CCD$ over the training steps when the highly regarded model CSQ [40] employs ResNet50 [9] or XnorNet [24] as backbones, with bit length $b = \{64, 32\}$. The results demonstrate that when CSQ

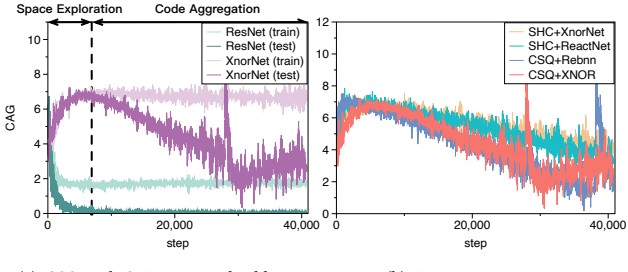

(a) CSQ with CNN or BNN backbone        (b) Various DH-BNN

**Figure 3: The Code AGgregation (CAG) analysis results on the IMAGENET100 dataset. A lower CAG implies a greater distance among hash codes with the same category. (a) CSQ employs ResNet50 or XnorNet as the backbone with bit length $b = 64$. (b) CSQ and SHC with various BNNs, the curve is about the train set.**

utilizes ResNet50 as the backbone, the hash codes rapidly converge toward their hash centers. In contrast, when XnorNet is used as the backbone, the hash codes initially approach the hash centers at a slower pace and later struggle to achieve closer alignment with the hash centers. Figure 2 (b) showcases the *CCD* results when CSQ employs other BNN backbones, including IR-Net [23], ReactNet [17], and Rebnn [34], which further substantiates this phenomenon. These findings highlight the limited representation capability of BNNs compared to CNNs [18], making it challenging to optimize the hash codes to reside in an ideal space.

From the hash code distribution perspective, center-based methods aim to ensure that hash codes of similar images are close to each other while dissimilar images are far apart. In fact, the optimization objective of center-based methods is precisely tailored to achieve this purpose. Considering that the hash centers can separate dissimilar hash codes, we analyze the aggregation of hash codes with the same category. We introduce the Codes AGgregation (CAG) metric defined as follows:

$$CAG = \frac{1}{m} \sum_{l=1}^{m} \frac{1}{|H^l|} \sum_{h_i \in H^l} D_H(h_i, \overline{h^l}). \qquad (6)$$

Here, $H^l$ represents the set of hash codes with category label $l$, and $\overline{h^l} = \frac{1}{|H^l|} \sum_{i \in H^l} h_i$ is the centroid of $H^l$. A lower value of *CAG* indicates a closer distance between hash codes of similar images. Figure 3 (a) depicts the evolution of *CAG* during the training steps when CSQ employs ResNet50 or XnorNet as backbones. When CSQ employs ResNet50 as its backbone, the hash codes with the same category progressively aggregate as the training step progresses. However, when CSQ utilizes XnorNet as its backbone, the *CAG* curve initially shows an increase followed by a limited decline. This phenomenon is further validated by various BNN-DH as shown in Figure 3 (b). We refer to the initial increase stage as the **space exploration stage** because it appears that BNNs initially explore the Hamming space. The subsequent decline stage is referred to as the **code aggregation stage** as the code attempts to aggregate together. Note that while a desired distribution of hash codes implies a lower *CAG*, a lower *CAG* does not necessarily guarantee a desired

distribution of hash codes. For instance, in an extreme scenario where all images share the same hash code, the *CAG* would be 0, but the search result is invalid. A similar situation can occur at the beginning of the space exploration stage. Therefore, it is advisable to consider the characteristics of the BNN-DH training process and selectively incorporate regularization methods to enhance the aggregation of hash codes with the same category.

## 4 THE PROPOSED ODH

In this section, we present a description of our proposed One-bit Deep Hashing (ODH) as depicted in Figure 4 to solve the above two problems. It consists of two main parts: a semantic self-adaptive hash center module and a two-stage training method.

### 4.1 Semantic Self-adaptive Hash Center Module

In our first analysis presented in Section 3.3, we find that hash codes generated by BNN-DH exhibit difficulties in attaining a close alignment with the hash centers. To tackle this problem, we propose a solution by transforming the hash center from a static pattern to a dynamic and adaptable mode, which simultaneously optimizes the hash codes to approach the hash center and the hash center to approach the hash codes.

As depicted in Figure 4 (a), we utilize the category label $y_i \in \{0, 1\}^m$ as input and aim to derive the hash center $c_i' \in \{-1, 1\}^b$. A straightforward approach involves inputting the label $y_i$ into a neural network, such as a Multilayer Perceptron (MLP), to obtain the hash center $c_i'$. However, this approach disregards the semantic information in the categories, leading to suboptimal results. Hence, we introduce a category feature matrix $W = [w_1, w_2, ..., w_m] \in R^{m \times C_m}$ to incorporate the semantic information of the categories, where $C_m$ is the dimension of $w_i$.

Taking inspiration from the previous work [29, 43], we devise a classifier task with the cross-entropy loss $L_{cls}$ as follows:

$$L_{cls} = -\frac{1}{N} \sum_{i=1}^{N} \sum_{l=1}^{m} y_{il} \log(\hat{y}_{il}), \qquad (7)$$

where $\hat{y}_i$ is the output of a CNN model (in our implementation, we utilize ResNet50, but other CNN models are also applicable) and $y_{il}, \hat{y}_{il}$ is the l-th value of $y_i, \hat{y}_i$ respectively. Then we can leverage the weights of the last fully connected layer in the classifier model as the category feature matrix $W$ as shown in the upper part of Figure 4 (a). Next, we perform a matrix multiplication between the one-hot encoded label $y_i = \{0, 1\}^m$ and the category matrix $W$, like a lookup process. This yields the category feature $w_i' = y_i W$ corresponding to the label $y_i$. Then, we employ a two-layer MLP with a sigmoid activation function $\sigma(\cdot)$ to map the category feature $w_i'$ to a representation $p_i \in (0, 1)^b$. The hash center is then generated by sampling from the multivariate Bernoulli distribution, resulting in $c_i' = Bernoulli(p_i)$. By employing the reparameterization trick [26], the entire process can be formulated as follows:

$$c_i' = \text{sign}(\sigma(\text{MLP}(y_i W^T)) - \mu), \qquad (8)$$

where $\mu$ denotes a sample drawn from the uniform distribution $U(0, 1)$. The gradient can be approximately estimated by applying the Straight-Through Estimator (STE) [1]. This approach enables the module to learn the binary output in an end-to-end fashion.

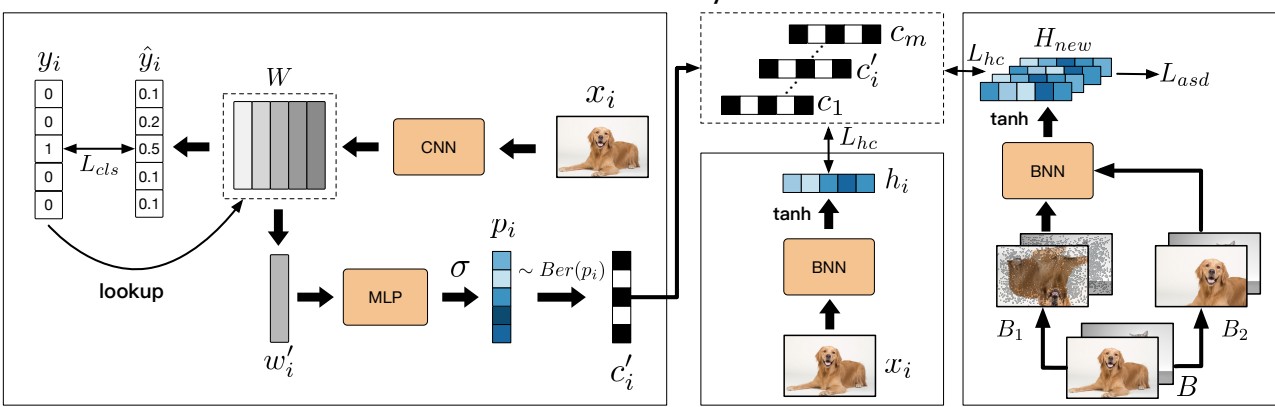

**(a) Semantic self-adaptive hash center module**    **(b) Two-stage training method**

Figure 4: Workflow of ODH. (a) The semantic self-adaptive hash center module. It generates hash centers with the category information. (b) The two-stage training method. (left) In the space exploration stage, ODH directly uses hash center loss $L_{hc}$ to force hash codes to approach hash centers. (right) In the code aggregation stage, ODH additionally utilizes a novel extrapolation aggregation objective $L_{asd}$ to enhance the aggregation of hash codes with the same category.

## 4.2 Two-stage Training Method

Once we obtain the hash centers $C' = \{c'_1, c'_2, ..., c'_N\}$, we can use them as the target to learn the hash model. However, based on the analysis in Section 3.3, BNN-DH usually exhibits two special stages: the space exploration stage and the code aggregation stage. To align with the changing characteristics of hash code aggregation in BNN-DH, we introduce a two-stage training method.

*4.2.1 Space Exploration Stage.* As shown in Figure 3 (a), during the space exploration stage, the CAG of BNN-DH gradually increases. This observation indicates that BNN-DH does not initially enhance the aggregation of hash codes with the same category. Based on this characteristic, as shown in the left of Figure 4 (b), we directly employ a BNN backbone with a $tanh(\cdot)$ function in the final layer to acquire the hash codes $h_i$ (In the inference stage, we replace $tanh(\cdot)$ to $sign(\cdot)$ function to get binary output). Then, we propose a hash center loss $L_{hc}$ to force the hash code $h_i$ to approach its hash center $c'_i$ while being far away from the hash centers of other categories. This can be achieved by maximizing the posterior probability of the ground-truth class using cross-entropy loss:

$$L_{hc} = -\frac{1}{N} \sum_i^N \log \frac{exp(\phi(h_i, c'_i)/\tau)}{\sum_{l=1}^m exp(\phi(h_i, c_l)/\tau)}. \qquad (9)$$

Here, $\phi(a_1, a_2) = \frac{a_1 a_2^T}{||a_1||_2 ||a_2||_2}$ represents the cosine similarity between $a_1$ and $a_2$. The notation $\tau$ denotes a temperature hyperparameter. Note that this optimization objective $L_{hc}$ can also promote a maximum distance between hash centers with weights $\frac{\mathcal{N}(l,k)}{b\tau Nm}$. $\mathcal{N}(l, k)$ is dependent on the number of samples per category present in the dataset. See Appendix A for details.

*4.2.2 Code Aggregation Stage.* During the code aggregation stage, in addition to employing the aforementioned center loss $L_{hc}$ to encourage hash codes to approach their hash centers, we propose

an extrapolation aggregation objective to enhance the aggregation among hash codes with the same category. The basic idea is to bring hash codes closer together within the same category in a training batch $B = \{x_i\}_{i=1}^{|B|}$, where $|B|$ is the batch size. However, focusing solely on the images in the training set may reduce the model's generalization capabilities. Another consideration is the importance of ensuring that images within the same category in a batch serve as hard samples for each other, thereby further promoting hash code aggregation. To achieve these goals, we propose an extrapolation aggregation objective $L_{asd}$. As shown in the right part of Figure 4 (b), we first utilize the data augmentation [12] to generate two views $B_1 = \{x_i^1\}_{i=1}^{|B|}$ and $B_2 = \{x_i^2\}_{i=1}^{|B|}$ for each image $x_i \in B$. Note that it is possible to extend this to more views, but this is not our primary focus. Assume $B_{new} = B_1 \cup B_2$ is the total images within a batch. $H_{new}$ is the corresponding hash code set. Then, considering the fact that increasing the variance between samples can facilitate the creation of hard samples [45], we propose a feature extrapolation method within the same category. Specifically, let $H_{new}^l$ represent the set of hash codes in $H_{new}$ that belong to category $l$. We compute the centroid vectors $S = \{s_i\}_{i=1}^r$, where $s_l = \frac{1}{|H_{new}^l|} \sum_{h_i \in H_{new}^l} h_i$ represents the centroid vector for each category in $H_{new}$. Here, $r$ indicates the number of $S$ and $|H_{new}^l|$ denotes number of $H_{new}^l$. As illustrated in Figure 5, for a hash code $h_i \in H_{new}^l$, we adopt weighted addition to generate a new one as follows:

$$\Phi(h_i) = \lambda h_i + (1 - \lambda)s_l. \qquad (10)$$

In this context, we select $\lambda$ from a uniform distribution as $\lambda \sim U(0, 1) + 1$, where the addition of 1 ensures a range of $(1, 2)$. This adjustment guarantees that $\hat{h}_i s_l \leq h_i s_l$, thereby increasing the variance of hash codes in $H_{new}^l$, as depicted in Figure 5 (b). Then,

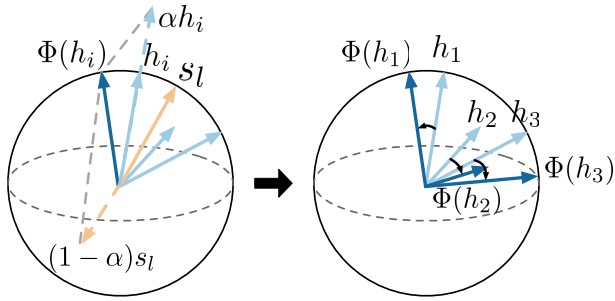

**Figure 5: The process of extrapolation used in the code aggregation stage. It aims to increase the variance among hash codes with the same category.**

the extrapolation aggregation objective is as follows:

$$L_{asd} = \sum_{l=1}^{r} \sum_{h_i, h_j \in H_{new}^l} D_H(\Phi(h_i), \Phi(h_j)). \qquad (11)$$

The objective Eq. 11 not only effectively harnesses the information from other images sharing the same category within a batch but also amplifies the variance of hash codes, leading to enhancement in code aggregation.

We transform the training method from the space exploration stage to the code aggregation stage after $T$ training epochs. We set $T$ as a hyper-parameter in our implementation, leaving the automated transition for future research. Moreover, in scenarios where the batch size $B$ is smaller than the number of categories $m$, some categories may have only one image in each training batch. To tackle this problem, we utilize a stratified sampling technique, which first samples some categories and then samples instances within these categories.

### 4.3 Training Procedure

Note that the dynamic hash centers are generated concurrently with the model's training. During the space exploration phase, we employ classification loss $L_{cls}$ and hash center loss $L_{hc}$ as targets for training our model as follows:

$$L = L_{hc} + \lambda_1 L_{cls}. \qquad (12)$$

After $T$ epochs, we enter the code aggregation stage and introduce the extrapolation aggregation loss $L_{sd}$, resulting in the final loss equation:

$$L = L_{hc} + \lambda_1 L_{cls} + \lambda_2 L_{asd}, \qquad (13)$$

where $\lambda_1$ and $\lambda_2$ are the hyper-parameters.

## 5 EXPERIMENTS

### 5.1 Experiment Settings

*5.1.1 Dataset.* We conduct extensive experiments on 11 deep hashing models based on two widely used image retrieval datasets. CIFAR100 [14] consists of 60,000 images from 100 classes. We randomly selected 50 images per class as the query set and 100 images per class as the training set and used all remaining images except queries as the database. IMAGENET100 is a subset of IMAGENET [4] with 100 classes. Following the settings from [40], we selected

100 categories and used all the images of these categories in the training set as the database and the images in the validation set as the queries. Furthermore, we randomly selected 13,000 as the training images from the database.

*5.1.2 Baselines and Training Details.* We considered the following deep hashing models as baselines from their learning objectives. Pair-wise methods include: DBDH [44], DCH [3], DFH [15], and DSHSD [32]. Ranking-based methods include: DTSH [31], TALR [8], and DTQ [16]. Center-based methods include: CSQ [40], DPN [6], SHC [29], and MDSH [30]. Besides, we considered the following binary neural networks as the backbones of deep hashing models: XnorNet [24], Bi2Real [20], IR-Net [23], ReactNet [17], Recu [37], Rbonn [35], and Rebnn [34].

We employed PyTorch to implement these models and conducted experiments on two Intel Xeon Gold 5218 CPUs and one NVIDIA Tesla V100. The batch size was set to 64, and we utilized the Adam optimizer [13] along with a grid search to select the learning rate from the set $\{10^{-1}, 10^{-2}, 10^{-3}, 10^{-4}, 10^{-5}\}$. Our model utilized the Adam optimizer for optimization, with a learning rate of $10^{-4}$. The hyperparameters $\lambda_1$, $\lambda_2$, and $\tau$ were set to 0.0001, 0.001, and 0.2 respectively. The value of $T$ was set to 100 for both datasets.

### 5.2 Performance on BNN-DH

In this experiment, we first used the mean Average Precision at the top K ($mAP@K$) as the evaluation metric to compare various deep hashing models utilizing different BNN backbones on CIFAR100 and IMAGENET100 datasets. Specifically, we adopted $mAP@500$ for IMAGENET100 and $mAP@1000$ for CIFAR100 while setting the code length $b = 64$. We also provide the result when these deep hashing models adopt full-precision backbone ResNet50.

Table 1 presents an overview of the results, revealing the following observations: (i) Our proposed ODH outperforms other baseline models significantly in various BNN backbones. Specifically, ODH surpasses the second-best result by 37.4% and 21.1% on the CIFAR100 and IMAGENET100 datasets, respectively, averaged across these BNN backbones. This indicates that ODH is a strong and general model. (ii) Overall, the center-based methods, such as CSQ, DPN, SHC, and MDSH, outperform the pair-wise methods and ranking-based methods in terms of performance. Some ranking-based methods and pair-wise methods, such as DFH and TALR, struggle to produce effective results. This suggests that the center-based objective is crucial when applying BNN to deep hashing models. We have a hypothesis to explain this phenomenon: BNNs have a smaller function space compared to CNNs, making it challenging to consider complex objectives. For instance, pair-wise and ranking-based methods need to consider at least $O(N^2)$ complex relationships, where $N$ is the number of images. The center-based method only needs to consider relationships ranging from $O(N)$ to $O(NM)$, where $M$ is the number of hash centers much smaller than $N$. Therefore, when using BNN as a backbone, the center-based method can achieve better and more stable results. (iii) ODH does not emerge as the foremost performer when utilizing the full-precision backbone ResNet because its design is based on analysis conducted with the BNN backbone. Furthermore, when using BNN

| Data | BNN | Pair-wise Models | | | | Ranking-based Models | | | Center-based Models | | | | |
|---|---|---|---|---|---|---|---|---|---|---|---|---|---|
| | | DBDH | DCH | DFH | DSHSD | DTSH | TALR | DTQ | CSQ | DPN | SHC | MDSH | ODH |
| CIFAR100 | ResNet | 0.407 | 0.567 | 0.449 | 0.575 | 0.526 | 0.452 | 0.514 | **0.646** | 0.643 | 0.638 | 0.628 | 0.607 |
| | XnorNet | 0.025 | 0.048 | 0.038 | 0.082 | 0.051 | 0.021 | 0.039 | 0.289 | 0.155 | 0.111 | 0.198 | **0.422***  |
| | Bi2Real | 0.037 | 0.188 | 0.019 | 0.147 | 0.151 | 0.056 | 0.158 | 0.218 | 0.250 | 0.317 | 0.145 | **0.421*** |
| | IR-Net | 0.208 | 0.179 | 0.029 | 0.201 | 0.188 | 0.063 | 0.204 | 0.310 | 0.276 | 0.166 | 0.127 | **0.448*** |
| | ReactNet | 0.115 | 0.267 | 0.017 | 0.211 | 0.292 | 0.091 | 0.235 | 0.335 | 0.296 | 0.328 | 0.301 | **0.471*** |
| | ReCu | 0.078 | 0.251 | 0.017 | 0.245 | 0.199 | 0.035 | 0.241 | 0.264 | 0.226 | 0.340 | 0.369 | **0.416*** |
| | Rbonn | 0.046 | 0.273 | 0.024 | 0.231 | 0.228 | 0.025 | 0.274 | 0.345 | 0.253 | 0.335 | 0.334 | **0.485*** |
| | Rebnn | 0.017 | 0.306 | 0.036 | 0.267 | 0.331 | 0.029 | 0.269 | 0.355 | 0.312 | 0.340 | 0.226 | **0.480*** |
| IMAGENET100 | ResNet | 0.533 | 0.859 | 0.562 | 0.856 | 0.784 | 0.591 | 0.797 | 0.877 | 0.870 | 0.894 | **0.895** | 0.861 |
| | XnorNet | 0.021 | 0.028 | 0.036 | 0.034 | 0.038 | 0.013 | 0.042 | 0.299 | 0.306 | 0.313 | 0.303 | **0.401*** |
| | Bi2Real | 0.025 | 0.188 | 0.019 | 0.128 | 0.133 | 0.026 | 0.174 | 0.349 | 0.330 | 0.324 | 0.265 | **0.361*** |
| | IR-Net | 0.112 | 0.197 | 0.030 | 0.241 | 0.191 | 0.042 | 0.235 | 0.330 | 0.349 | 0.329 | 0.303 | **0.455*** |
| | ReactNet | 0.069 | 0.326 | 0.019 | 0.262 | 0.286 | 0.044 | 0.289 | 0.362 | 0.390 | 0.423 | 0.377 | **0.501*** |
| | ReCu | 0.091 | 0.230 | 0.019 | 0.286 | 0.203 | 0.031 | 0.216 | 0.209 | 0.243 | 0.257 | 0.249 | **0.314*** |
| | Rbonn | 0.036 | 0.316 | 0.027 | 0.153 | 0.268 | 0.127 | 0.247 | 0.418 | 0.432 | 0.419 | 0.399 | **0.508*** |
| | Rebnn | 0.019 | 0.303 | 0.033 | 0.113 | 0.240 | 0.049 | 0.238 | 0.386 | 0.404 | 0.374 | 0.381 | **0.515*** |

**Table 1: The mAP@K comparison results on IMAGENET100 and CIFAR100 when using different hashing models and BNN backbones. Code length $b = 64$. The best result in each column is marked in bold. The second-best result in each column is underlined. ∗ represents statistically significant improvements over the best baseline with $p < 0.05$ using a two-tailed paired t-test. This notation is also used in Table 2.**

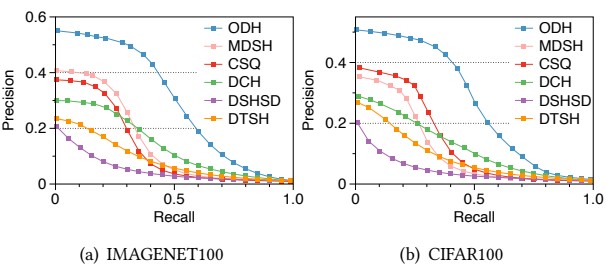

**(a) IMAGENET100**      **(b) CIFAR100**

**Figure 6: The Precision-Recall curve comparison on two datasets for five representative baselines.**

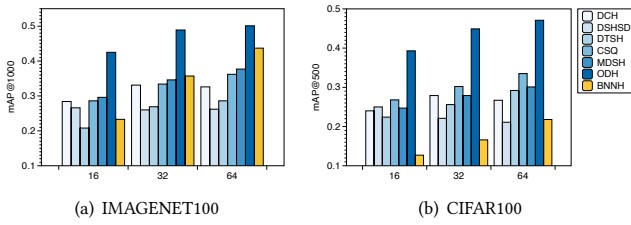

**(a) IMAGENET100**      **(b) CIFAR100**

**Figure 7: The *mAP@K* comparison when set different code lengths $b \in \{16, 32, 64\}$.**

as the backbone, the hardware utilized significantly influences practical computation costs. Consequently, we refer to [18] and offer a theoretical efficiency analysis in Appendix B.

Besides, we conducted more comprehensive comparisons between ODH and the baselines to evaluate their performance from various aspects. Specifically, in subsequent experiments, we selected DCH, DSHSD, DTSH, CSQ, and MDSH as baseline, which achieved good performance in the previous experiment. ReactNet was used as the backbone. First, we calculated the Precision-Recall curves for comparison. Figure 6 illustrates the results of these models on the IMAGENET100 and CIFAR100 datasets when the bit length $b = 64$. It is evident that ODH consistently attains superior results compared to all other baselines, particularly when the recall is relatively lower. As the recall gradually increases, the precision of ODH slowly diminishes, whereas some baselines witness a rapid decline in precision. These observations further emphasize the effectiveness of ODH. Second, we compared their performance at different bit lengths. In addition to the baseline models selected for the previous experiments, we also included BNNH [41] for performance comparison[1]. As illustrated in Figure 7, our ODH model outperforms other baselines under the same bit length. Besides, we can find that BNNH does not consistently achieve better performance compared to other baselines, especially in the CIFAR100 dataset. This highlights the importance of designing BNN-agnostic deep hashing, which can adopt a powerful BNN backbone like ReactNet.

## 5.3 Model Analysis

In this experiment, we conduct a comprehensive analysis of ODH from multiple perspectives, including (i) Ablation study, (ii) Parameter analysis, and (iii) CCD and CAG analysis.

*5.3.1 Ablation Study.* To investigate the impact of different components in ODH, we devised several variants, namely (i) ODH w/o S: ODH without utilizing the semantic self-adaptive hash center module, employing a fixed hash center similar to CSQ [40]. (ii) ODH w/o E: ODH without extrapolation aggregation objective in

---

[1]We use the modified BNN-VGG16 architecture as described in their paper

**Table 2: The $mAP@K$ results for different ODH variants on two datasets and set code length $b \in \{16, 32, 64\}$.**

| Model | IMAGENET100 | | | CIFAR100 | | |
|---|---|---|---|---|---|---|
| | 16bit | 32bit | 64bit | 16bit | 32bit | 64bit |
| ODH w/o S | 0.301 | 0.335 | 0.357 | 0.324 | 0.331 | 0.353 |
| ODH w/o E | 0.324 | 0.351 | 0.364 | 0.336 | 0.344 | 0.365 |
| ODH w/o T | 0.345 | 0.371 | 0.384 | 0.361 | 0.376 | 0.407 |
| ODH | **0.368*** | **0.389*** | **0.401*** | **0.381*** | **0.401*** | **0.422*** |

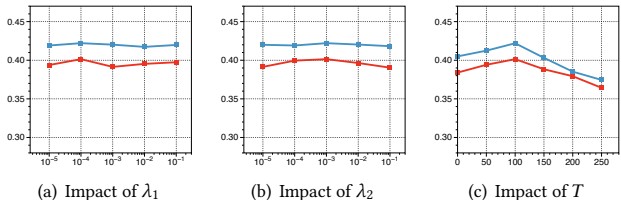

(a) Impact of $\lambda_1$     (b) Impact of $\lambda_2$     (c) Impact of $T$

**Figure 8: The $mAP@K$ performance when hyper-parameter $\lambda_1$, $\lambda_2$, and $T$ set different values. Red line denotes IMA-GENET100 dataset. Blue line denotes CIFAR100 dataset.**

the code aggregation stage. (iii) ODH w/o T: ODH directly skips the space exploration stage. We set the code length $b = 64$ and employ XnorNet [24] as the backbone. The results are presented in Table 2, where we can derive the following conclusions. First, through the comparison of ODH with ODH w/o S, we observe a notable decline in performance. This signifies the effectiveness of the semantic self-adaptive hash center module. Second, we note that ODH outperforms ODH w/o E, indicating that adopting the extrapolation aggregation objective yields better results than direct training. Furthermore, comparing ODH with ODH w/o T can demonstrate that the two-stage training approach necessary.

*5.3.2 Parameter Analysis.* To examine the impact of the key hyper-parameters of ODH on performance, we evaluated the $mAP@K$ when ODH applies different $\lambda_1$, $\lambda_2$, and $T$. First, the parameters $\lambda_1$ and $\lambda_2$ in Eq. (13) can be used to adjust the weighting of $L_{cls}$ and $L_{asd}$. We set their values from $\{10^{-1}, 10^{-2}, 10^{-3}, 10^{-4}, 10^{-5}\}$ to explore different combinations. Besides, We set the bit length $b = 64$ and utilized the XnorNet as the backbone. Figure 8 depicts the results, where we can find that ODH is not sensitive to the settings of these two parameters. Second, the parameter $T$ controls the timing of transitioning from the space exploration stage to the code aggregation stage. Thus we set $T = \{0, 50, 100, 150, 200, 250\}$. As depicted in Figure 8 (c), we can observe that $T$ is not very sensitive to the results when it is in the middle range. However, when $T$ is set too large, it may obviously impact the results.

*5.3.3 CCD and CAG analysis.* Codes Center Distance (CCD) and Codes AGgregation (CAG) are two important metrics we introduced in Section 3.3. We chose the top 10 minimum CCD and CAG values that have occurred during the training process. Then, we calculated their mean as the final result (we used the test set to calculate CCD and CAG). The heatmap displayed in Figure 9 presents the results for CCD. Across various BNN backbones, it is evident that ODH

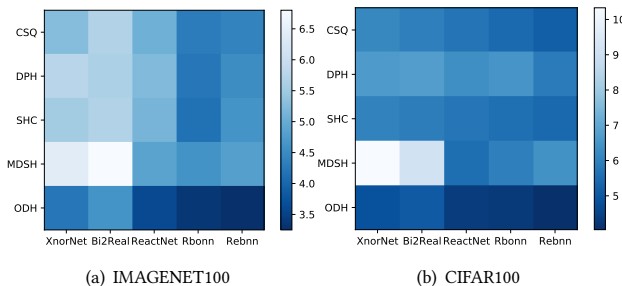

(a) IMAGENET100        (b) CIFAR100

**Figure 9: The Code Center Distance (CCD) analysis. Darker colors indicate lower CCD values, indicating that the hash codes are closer to their hash centers.**

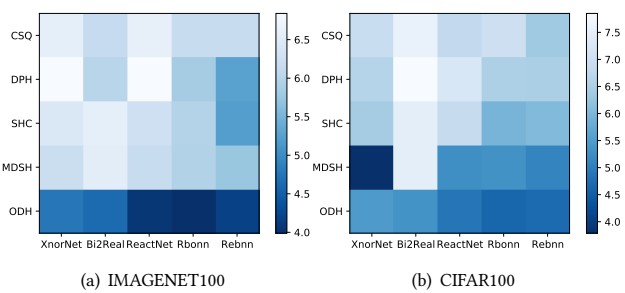

(a) IMAGENET100        (b) CIFAR100

**Figure 10: The Code AGgregation (CAG) analysis. Darker colors indicate lower CAG values, indicating a closer distance between hash codes of similar images.**

achieves lower CCD values compared to other deep hashing models. The consistently smaller CCD values indicate that our approach is better suited for center-based learning objectives. This can be attributed to the utilization of the semantic self-adaptive hash center module, which not only encourages hash codes to approach hash centers but also ensures that hash centers move closer to the hash codes. Figure 10 displays the results for CAG. ODH achieves a lower CAG compared to most deep hashing models, indicating its effectiveness in enhancing the aggregation of hash codes with the same category.

## 6 CONCLUSION

In this paper, we delved into the practical challenge of implementing deep hashing models on devices with limited resources. We merged binary neural networks with deep hashing models to address this issue. Through comprehensive analyses, we proposed a novel approach called One-bit Deep Hashing (ODH), which incorporates a semantic self-adaptive hash center module and a novel two-stage training method. We conducted extensive experiments on two datasets to evaluate the performance of ODH. The results demonstrate the superiority of ODH and establish it as a foundational framework for future research in binarized deep hashing models.

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
