# OpenReview forum: "One-bit Semantic Hashing: Towards Resource-Efficient Hashing Model with Binary Neural Network"
_acmmm.org/ACMMM/2024/Conference — MM2024 Poster_

### Official Review · Reviewer_Z4Mj · 2024-05-21

**Rating:** 3
**Confidence:** 3

**Summary:**

To deploy Deep Hashing (DH) models on resource-constrained devices, previous work has proposed using Binary Neural Networks (BNN) instead of the original Convolutional Neural Networks (CNN) to reduce computations and parameters. However, directly applying BNN to DH results in hash codes that struggle to converge to their respective hash centers, and hash codes of images from the same category also find it difficult to aggregate together, ultimately diminishing the performance of image retrieval tasks. In order to solve these problems, this paper proposes a novel model, One-bit Deep Hashing (ODH), which combines BNN and DH into a unified framework. This model incorporates a semantic self-adaptive hash center module and a novel two-stage training method. Extensive experiments on two widely used datasets demonstrate that ODH can achieve significant superiority over other BNN-DH models.

**Strengths:**

(1)The paper designs a module capable of dynamically generating hash centers, which simultaneously optimizes the hash codes to approach the hash center and the hash center to approach the hash codes. Additionally, a two-stage training method is proposed to align with the changing characteristics of code aggregation in BNN-DH.

(2)The paper is well-organized, making it easy for readers to follow the logical flow of the research. Additionally, the use of figures and tables is strategic, enhancing comprehension without overwhelming the reader.

**Limitations:**

(1)There might be an issue with Equation (3) in the paper. First, the upper bound of the summation should be $b$. Moreover, the Hamming distance between  $h_i$ and $h_j$ should be $b-POPCOUNT(h_i\ XNOR\ h_j)$ , whereas Equation (3) calculates the number of identical bits between the two hash codes. In addition, the $\lambda$ in Equation (10) and the $\alpha$ in Figure (5)a are not inconsistent. Clear and well-defined symbol usage is essential for effective communication of scientific ideas and should be prioritized in this paper.

(2)The preliminary analysis mentioned that during the training of center-based BNN models, the CAG metric initially increases and then decreases, affecting the aggregation of hash codes within the same category. Therefore, would the issue be mitigated by the two-stage training method you designed? Because in the code aggregation stage, the paper also increases the distance between hash codes of the same category. Moreover, the experiments only compared the top 10 minimum CAG values during training and did not demonstrate the impact of the method on the initial CAG values.

**Suitability:**

2

---

### Official Review · Reviewer_3Y3S · 2024-05-22

**Rating:** 4
**Confidence:** 4

**Summary:**

This manuscript proposes a method (ODH) that integrates BNNs with deep hashing to create a resource-efficient hashing model. ODH introduces a semantic self-adaptive hash center module to address issues with hash code convergence and employs a two-stage training method to enhance hash code aggregation. Experiments have been conducted on two datasets.

**Strengths:**

1. This manuscript provides a thorough theoretical analysis of integrating BNNs with center-based methods and two-stage aggregation.
2. This manuscript has a clear motivation, i.e., combining BNNs and deep hashing into a unified framework to overcome challenges for deploying deep hashing on resource-limited devices.
3. This manuscript is well-organized and clearly written.

**Limitations:**

1. The method design is flawed and confusing. At the Code Aggregation Stage, **using CNN**, e.g., ResNet50, to train dynamic hash centers in a resource-constrained environment seems to violate the paradigm of this article. Additionally, while using data augmentation methods to facilitate the aggregation of hash codes belonging to the same category is interesting, but not novel.
2. There is limited discussion on the scalability of the proposed method when applied to multi-label datasets like MIRFLICKR or higher-dimensional hash codes, e.g., 128-bit. Relevant experimental results should be provided.
3. What is the reason that CNN-based deep hashing methods do not exhibit the space exploration phase (phenomenon) similar to BNN-DH?
4. There are presentation issues, and further proofreading is highly needed. In section 3.1, the Hamming distance is usually measured by the XOR operation.
5. The CCD analysis and CAG analysis are provided in section 5.3.3. I am wondering, in the space exploration stage you defined, how about the trend of CAG? Is it possible to achieve a descending effect directly like CNN-DH?

**Suitability:**

2

---

### Official Review · Reviewer_UyJv · 2024-05-22

**Rating:** 4
**Confidence:** 3

**Summary:**

This paper proposes a novel deep hashing method using Binary Neural Networks (BNNs), named One-bit Deep Hashing (ODH). By applying BNNs directly to Deep hashing, the authors first discovered two challenges stemming from the limited representation capability of BNNs and the two-stage hash code aggregation in BNN-DH. To tackle these challenges, the authors first introduce a semantic self-adaptive hash center module to facilitate the coverage of hash codes. Additionally, the authors propose a two-stage training strategy in the hash code aggregation period. Extensive experiments evaluate the effectiveness of the proposed method.

**Strengths:**

1. The CCD and the CAG analysis illustrate the challenges of center-based methods when using BNN from two different perspectives.
2. The paper provides a comprehensive overview of deep hashing and binary neural networks, offering valuable context for the proposed method.
3. A novel semantic self-adaptive hash center module is introduced, which facilitates the coverage of hash codes.
4. The experiments conducted are extensive, and the results are presented with clarity, supporting the validity of the proposed approach.

**Limitations:**

1. The proposed components are general in both BNN-based and CNN-based DH methods. The authors should provide more explanation regarding the design of the ODH method specifically for BNNs. **Specifically**, they should clarify why space exploration issues arise when using BNNs and how the two-stage training strategy addresses these issues effectively.

2. In Figures 2 and 3, the CCD decreases when using BNNs, indicating that the hash codes are becoming closer to the fixed center. However, the CAG initially increases, suggesting larger distances between hash codes within the same classes. Why do these two metrics show opposite trends?

3. The caption of Figure 3 should change to "A lower CAG implies a ~~greater~~ **smaller**  distance among hash codes with the same category."

4. Figure 10 shows that MDSH exhibits lower CAG values with XnorNet on CIFAR100 than ODH, but the mAP results of MDSH are lower, why?

5. The variable $\hat{h}_i$ is not pre-defined before its use.

6. There are several inconsistent expressions of the extrapolation aggregation loss like $L_{a}sd$, $L_{sd}$, and $L_{asd}$.

**Suitability:**

2

---

### Official Review · Reviewer_RgVe · 2024-05-27

**Rating:** 4
**Confidence:** 2

**Summary:**

This paper works on the binarization of deep hashing models. A One-bit Deep Hashing (ODH) method is proposed, which incorporates
a semantic self-adaptive hash center module and a novel two-stage training method. Extensive experiments show the superiority of ODH over other methods.

**Strengths:**

1. The proposed dynamic hash center method is interesting and seems effective.
2. Extensive comparisons and ablation studies are provided.

**Limitations:**

1. I didn't see the strong correlation between the BNN and the proposed method. It seems that the proposed method should also works for ResNet-50.
2. In Figure 3, why space exploration and code aggregation only occur for BNN but not for ResNet-50? What's the hyper-parameters to train the deep hashing models with BNN backbones?
3. Could the authors provide results using other light-weight models, e.g., sparse models, or MobileNets, etc.

**Suitability:**

3

---

### Meta-Review · Area_Chair_RWRF · 2024-06-29

**Recommendation:** Accept (Poster)
**Confidence:** 5

**Metareview:**

This study discovers issues in applying BNN to DH, reveals key insights, designs the One-bit Deep Hashing (ODH) method with a hash center module and two-stage training, and shows its superiority through experiments on two datasets. Overall, the paper is technically solid and will have high impact on multimodal perception research. Given the resolution of raised concerns and the unanimous positive reviews, the paper is accepted for publication.